# UOE: Unlearning one Expert is Enough for Mixture-of-Experts LLMs

## Abstract

Recent advancements in large language model (LLM) unlearning have shown remarkable success in removing unwanted data-model influences while preserving the model's utility for legitimate knowledge. However, despite these strides, sparse Mixture-of-Experts (MoE) LLMs–a key subset of the LLM family–have received little attention and remain largely unexplored in the context of unlearning. As MoE LLMs are celebrated for their exceptional performance and highly efficient inference processes, we ask: How can unlearning be performed effectively and efficiently on MoE LLMs? And will traditional unlearning methods be applicable to MoE architectures? Our pilot study shows that the dynamic routing nature of MoE LLMs introduces unique challenges, leading to substantial utility drops when existing unlearning methods are applied. Specifically, unlearning disrupts the router's expert selection, causing significant selection shift from the most unlearning target-related experts to irrelevant ones. As a result, more experts than necessary are affected, leading to excessive forgetting and loss of control over which knowledge is erased. To address this, we propose a novel single-expert unlearning framework, referred to as UOE, for MoE LLMs. Through expert attribution, unlearning is concentrated on the most actively engaged expert for the specified knowledge. Concurrently, an anchor loss is applied to the router to stabilize the active state of this targeted expert, ensuring focused and controlled unlearning that preserves model utility. The proposed UOE framework is also compatible with various unlearning algorithms. Extensive experiments demonstrate that UOE enhances both forget quality up to $5\%$ and model utility by $35\%$ on MoE LLMs across various benchmarks, LLM architectures, while only unlearning $0.06\%$ of the model parameters.

## 1 Introduction

Despite the extraordinary ability in generating human-like content (Touvron et al., 2023), the rapid development of large language models (LLMs) have raised a series of ethical and security concerns, such as pretraining on copyrighted data (Sun et al., 2024), bias perpetuation (Motoki et al., 2023), the generation of toxic, biased, or illegal contents (Wen et al., 2023), and facilitating making cyber-attacks and bio-weapons (Li et al., 2024). As a solution, the problem of Machine Unlearning (MU) arises (also referred to LLM unlearning) (Liu et al., 2024c), aiming to scrub the influence of the undesired training data and removing their corresponding generation abilities, while preserving the influence of other remaining valid data (Jia et al., 2024a; Shi et al., 2024; Jia et al., 2024b).

While LLM unlearning has recently become a major research thrust, past efforts have only focused on effective unlearning methods for conventional LLMs. In contrast, sparse Mixture-of-Experts LLM (MoE LLM) (Jiang et al., 2024; xAI, 2024; Databricks, 2024; Abdin et al., 2024; Liu et al., 2024a), designed to reduce computational burdens during inference, remained unexplored in this context. As a key member of the LLM family, MoE LLMs offer substantial scalability without a corresponding increase in computational costs (Jiang et al., 2024; Team, 2024; Dai et al., 2024). Thanks to their dynamic routing mechanism, MoE LLMs direct inference through different model components, known as 'experts'. However, it remains unclear how LLM unlearning interacts with the sparse MoE architecture and whether unlearning for MoE LLMs presents unique challenges. This leads us to ask:

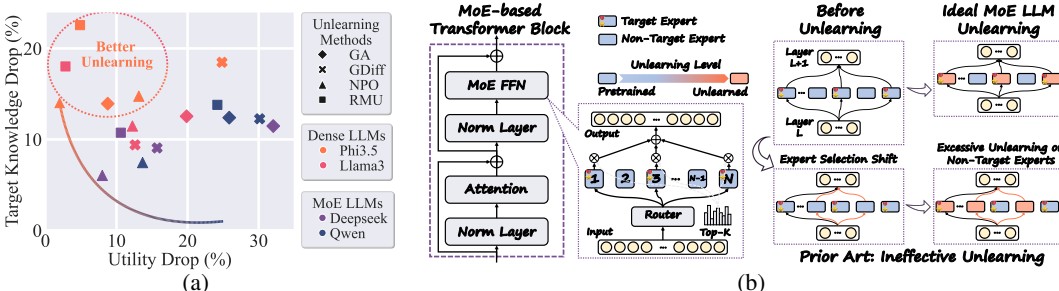

Figure 1: Overview of the key findings in this paper. (a) Illustration of the ineffectiveness of existing unlearning methods on MoE LLMs. Four unlearning algorithms—GA (Eldan & Russinovich, 2023), GDIFF (Maini et al., 2024), NPO (Zhang et al., 2024), and RMU (Li et al., 2024)—were applied to two MoE LLMs (DeepSeek-v2-Lite (Liu et al., 2024a) and Qwen1.5-MoE (Team, 2024)) and two dense LLMs (Phi3.5 (Abdin et al., 2024) and LLaMA3-8B (Dubey et al., 2024)) using the WMDP benchmark Li et al. (2024). The drop in target knowledge (accuracy drop on the forget test set, higher is better) and the drop in model utility (accuracy drop on MMLU Hendrycks et al. (2023), lower is better) are plotted. Ideal performance is in the top left corner, but MoE LLMs show poor unlearning quality with sharp utility drop. (b) Illustration of ideal versus ineffective MoE LLM unlearning. Target experts—those most frequently activated given the forget set—are identified for unlearning. However, existing unlearning algorithms tend to cause substantial expert selection shifts, leading to excessive and unnecessary unlearning of non-target experts, which significantly impairs model utility.

> *(Q) Can we develop a principled MU method for MoE LLMs that ensures high forgetting effectiveness, while maintaining model utility and efficiency?*

To the best of our knowledge, the problem (**Q**) remains unexplored in the current literature. Our investigation begins with a pilot study that applies existing unlearning methods to MoE LLMs. Preliminary results indicate that a simple implementation of these methods can lead to a substantial drop in model utility and even model collapse. This phenomenon is illustrated in **Fig. 1**(a), which depicts the performance of the unlearned MoE LLMs predominantly closer to the bottom right corner, indicating a significant and unacceptable utility drop compared to conventional dense LLMs.

To fully understand this phenomenon, we begin by performing a careful sanity check on unlearning methods in MoE LLMs and conduct an in-depth analysis of failure cases. Ideally, in MoE LLMs, given an input, the routers should evaluate and direct it to the most relevant experts, with unlearning targeting and erasing the corresponding knowledge in these experts. However, by monitoring expert selection during unlearning, we find that the process often prompts routers to constantly switch the activated experts. This behavior persists even when routers are fixed. As a result, unlearning algorithms create "short-cuts", where instead of targeting the most relevant experts, the routers shift to less relevant ones to trick for unlearning loss reduction (*i.e.*, irrelevant experts could be unlearned). This leads to substantial drops in model utility. See **Fig. 1**(b) for illustration.

To solve the problem, we propose a novel unlearning framework specifically tailored for MoE LLMs, named UOE, which stands for Unlearning One Experts. UOE employs expert attribution to pinpoint the expert most actively involved with the forget set, which is designated as the primary target for unlearning. Unlearning efforts are exclusively focused on this identified expert. Concurrently, an anchor loss is applied to the router to stabilize the active status of the targeted expert throughout the unlearning process. This approach prevents the frequent switching of expert selection, ensuring that unlearning is both focused and controlled. Our contributions are summarized below.

- We for the first time identify the unique challenge of unlearning in MoE LLMs. Our analysis elucidates the root causes of observed failures, offering novel insights into how unlearning impacts the routers and experts within an MoE LLM.

- We propose a novel parameter-efficient unlearning framework, UOE, tailored for MoE LLMs. UOE effectively pinpoints, fixates, and unlearns the most pertinent expert relative to the forget set. UOE enjoys high flexibility and works in a plug-in-and-play mode with any existing unlearning methods to boost forget quality, model utility, and efficiency at the same time.

- We conduct extensive experiments to demonstrate the effectiveness of UOE across various MoE architectures, MU benchmarks, and unlearning methods. Our results show that when integrated with UOE, all tested unlearning methods achieve significant improvements in model utility up to 35% and concurrently enhance the quality of forgetting with only 0.06% parameters being updated.

## 2 RELATED WORKS

**Machine Unlearning for LLMs.** A growing body of research has investigated the problem of unlearning in large language models (LLMs) (Yao et al., 2024; Lu et al., 2022; Jang et al., 2022; Kumar et al., 2022; Zhang et al., 2023a; Pawelczyk et al., 2023; Eldan & Russinovich, 2023; Ishibashi & Shimodaira, 2023; Yao et al., 2023; Maini et al., 2024; Zhang et al., 2024; Li et al., 2024; Wang et al., 2024a; Jia et al., 2024b; Liu et al., 2024c;b; Thaker et al., 2024). These studies have practical applications, such as removing sensitive information (Jang et al., 2022; Eldan & Russinovich, 2023; Wu et al., 2023) and preventing the generation of harmful or biased content (Jang et al., 2022; Eldan & Russinovich, 2023; Wu et al., 2023; Lu et al., 2022; Yu et al., 2023; Yao et al., 2023; Liu et al., 2024d), memorized sequences (Jang et al., 2022; Barbulescu & Triantafillou, 2024), and copyrighted material (Eldan & Russinovich, 2023; Jang et al., 2022). To facilitate unlearning, recent methods aim to bypass the need for retraining models from scratch by excluding the forget set containing the targeted data to be removed (Ilharco et al., 2022; Liu et al., 2022; Yao et al., 2023; Eldan & Russinovich, 2023; Jia et al., 2024b; Zhang et al., 2024; Li et al., 2024; Thaker et al., 2024; Liu et al., 2024b). Techniques like task arithmetic also enable efficient model editing through parameter merging (Hu et al., 2024; Ilharco et al., 2022). Although these methods do not provide exact unlearning akin to full retraining, they remain efficient and effective under empirical unlearning evaluation metrics. Approaches often include model fine-tuning and optimization (Liu et al., 2022; Yao et al., 2023; Eldan & Russinovich, 2023; Jia et al., 2024b; Zhang et al., 2024; Li et al., 2024), or input prompting and in-context learning (Thaker et al., 2024; Pawelczyk et al., 2023; Liu et al., 2024b). Other approaches, such as localization-informed unlearning, identify and locally edit model units (e.g., layers or neurons) closely related to the data or tasks being unlearned (Meng et al., 2022; Wu et al., 2023; Wei et al., 2024). Despite these efforts, studies have shown that forgotten knowledge can often still be extracted from models post-unlearning (Patil et al., 2024; Liu et al., 2024e; Lynch et al., 2024; Shostack, 2024). However, most existing research has focused on dense LLMs, leaving unlearning in MoE LLMs largely unexplored. For example, the unlearning of Mixtral-$8 \times 7B$ is discussed in Li et al. (2024), but only a single method with ad-hoc adjustments was examined. This work aims to fill this gap by conducting a comprehensive study of various unlearning methods, benchmarks, and MoE models, addressing the specific challenges posed by the MoE architecture.

**MoE-based LLMs.** Sparse Mixture-of-Experts (MoE) models are designed to activate only a subset of expert networks for each input, enabling substantial model scaling with minimal computational overhead (Shazeer et al., 2017). Current approaches to MoE model development can be categorized into two types: training from scratch (Fedus et al., 2022; Zoph et al., 2022a; Shen et al., 2023) and building from dense checkpoints (Zhang et al., 2021; Komatsuzaki et al., 2022; Zhu et al., 2024). Over recent years, MoE models have seen key advancements, including improvements in scalability (Riquelme et al., 2021; Kim et al., 2021; Zhou et al., 2022; Zoph et al., 2022a), efficiency optimization (Fedus et al., 2022; Lepikhin et al., 2020; Chowdhery et al., 2023), and expert balancing techniques (Cong et al., 2024; Zoph et al., 2022b; Dai et al., 2022). The implementation of transformer-based MoE models has been successfully integrated into LLMs, significantly enhancing inference efficiency (Jiang et al., 2024; Dai et al., 2024; Team, 2024; xAI, 2024; Hong et al., 2024; Abdin et al., 2024; Lieber et al., 2024; Yang et al., 2024; Zhu et al., 2024; Databricks, 2024; Xue et al., 2024). For example, DeepSeekMoE (Dai et al., 2024) improves expert specialization by segmenting experts into smaller subsets for flexible activation, while isolating shared experts to reduce redundancy and capture common knowledge. Similarly, Qwen1.5-MoE (Team, 2024) partitions a standard FFN layer into smaller segments to create multiple experts, introducing a fine-grained routing mechanism that enables Qwen1.5-MoE to match the performance of 7B models while using only one-third of the activation parameters. Despite the efficiency gains provided by MoE's dynamic routing system, existing research highlights additional challenges compared to traditional dense models, including unstable training (Zoph et al., 2022a; Dai et al., 2022), robustness issues (Zhang et al., 2023b; Puigcerver et al., 2022), and complications in parallel deployment (Hwang et al., 2023; Gale et al., 2023). In this work, we show that the root cause of the ineffectiveness of existing unlearning methods for MoE LLMs also stems from the dynamic routing system.

## 3 PRELIMINARIES

In this section, we start by presenting the mathematical formulation of LLM unlearning. The lack of exploration on MoE LLM unlearning inspires us to investigate whether existing unlearning methods keep effective in these models. Our pilot study reveals that methods designed for conventional LLMs are *ineffective* in unlearning MoE LLMs.

**Preliminaries on MoE LLM unlearning.** Based on the generic formulation of LLM unlearning outlined in Liu et al. (2024c), the task of LLM unlearning can be formulated as eliminating the influence of a specific 'unlearning target'–whether it is related to data, knowledge, or model capabilities–from a pretrained LLM (denoted by $\boldsymbol{\theta}_o$). The unlearning target is typically defined by a forget set $\mathcal{D}_f$, which contains the information or knowledge to be removed. To ensure the model retains its generation ability (*i.e.*, utility) after unlearning, a retain set $\mathcal{D}_r$ is introduced, consisting of data unrelated to the unlearning target. With this setup, the LLM unlearning problem is usually formed as a regularized optimization problem, finetuned from $\boldsymbol{\theta}_o$ using both the forget set $\mathcal{D}_f$ and the retain set $\mathcal{D}_r$:

$$\min_{\boldsymbol{\theta}} \ell_f(\boldsymbol{\theta}; \mathcal{D}_f) + \lambda \ell_r(\boldsymbol{\theta}; \mathcal{D}_r). \tag{1}$$

Here, $\boldsymbol{\theta}$ represents the model parameters to be updated during unlearning, $\ell_f$ and $\ell_r$ denote the forget loss and retain loss, respectively, with $\lambda \geq 0$ serving as a regularization parameter to balance between unlearning and preserving utility.

Next, we provide a brief introduction to how the routing system operates in the MoE LLM architecture. In MoE LLMs, *e.g.*, DeepSeek-v2-Lite (Liu et al., 2024a), the feed-forward networks (FFNs) of Transformers are split into multiple experts and activated by the output of the router in front of the expert layers, see Fig. 1(b) for illustration. In the $l$-th layer, given the input $\mathbf{u}_t^{(l)}$ corresponding to the $t$-th token, router layers calculate the score of each token and assign them to top-$K$ experts:

$$s_{i,t}^{(l)} = \text{Softmax}(\text{Router}(\mathbf{u}_t^{(l)}))$$

$$g_{i,t}^{(l)} = \begin{cases} s_{i,t}^{(l)} & \text{if } s_{i,t}^{(l)} \in \text{Top}K(\{s_{k,t}^{(l)} \mid 1 \leq k \leq N\}) \\ 0 & \text{otherwise} \end{cases}$$

Here, Router$(\cdot)$ denotes the router layer, $s_{i,t}$ is the token-to-expert affinity, Top$K(\cdot)$ selects the highest $K$ value in the set, $N$ is the number of experts, and $g_{i,t}^{(l)}$ is the score assigned by router for the $i$-th expert. Then, the hidden state $\mathbf{h}_t'^{(l)}$ of FFNs can be calculated as: $\mathbf{h}_t'^{(l)} = \mathbf{u}_t^{(l)} + \sum_{i=1}^{N} g_{i,t}^{(l)} \text{FFN}_i^{(l)}(\mathbf{u}_t)$, where $\text{FFN}_i^{(l)}(\cdot)$ denotes the $i$-th expert. Then, $\mathbf{h}_t'^{(l)}$ is sent to the next layer of Transformer blocks for further processing.

**Unlearning for MoE LLM is not trivial: a pilot study.** The goal of unlearning is twofold: (1) to ensure the model forgets the targeted information and knowledge stored in $\mathcal{D}_f$, and (2) to preserve the model utility without significant degradation. Our pilot study reveals that the special routing system in MoE LLMs introduces additional challenges to unlearning, rendering existing methods ineffective. We applied four widely used LLM unlearning methods: GA (Gradient Ascent) (Eldan & Russinovich, 2023), GDIFF (Gradient Difference) (Maini et al., 2024), NPO (Negative Preference Optimization) (Zhang et al., 2024), and RMU (Representation Misdirection for Unlearning) (Li et al., 2024) with the WMDP benchmark (Li et al., 2024) on two MoE LLMs, Qwen1.5-MoE (Team, 2024) and DeepSeek-V2-Lite (Liu et al., 2024a), as well as two dense LLMs for reference, LLaMA3-8B (Dubey et al., 2024) and Phi-3.5-mini-instruct (Abdin et al., 2024), where the task aims to unlearn hazardous knowledge in LLMs. In **Fig. 1(a)**, to ease the comparison, we report the forget quality (performance drop on the forget test set, where higher is better) against retain quality (performance drop on the MMLU (Hendrycks et al., 2020) utility benchmark, where lower is better). Each data point represents the best result of a model-method combination with hyper-parameter tuning, with ideal performance located near the top left corner, signifying high unlearning effectiveness with minimal impact on model utility. As we can see, most MoE LLM data points cluster in the lower right, indicating severe utility drops and poor unlearning performance compared to dense models. In Fig. 1(a), all model parameters (including routers and experts) are involved in unlearning. To ensure that these poor results are not due to improper parameter settings, **Tab. 1** presents additional experiments using two other parameter configurations (routers-only and experts-only) for GA, yet no significant improvements are observed in either forget or retain quality (more than 20% utility drop). The results above imply the problem of MoE LLM unlearning is more challenging and far from trivial, even if LLM unlearning is well-studied.

Table 1: Unlearning performance when controlling tunable parameters in MoE LLMs.

| Tunable Module | Forget Quality ↓ | Retain Quality ↑ |
|---|---|---|
| Qwen | | |
| Original | 0.4192 | 0.5979 |
| Experts & Router | 0.2953 | 0.3393 |
| Routers Only | 0.2526 | 0.2977 |
| Experts Only | 0.2536 | 0.3242 |
| DeepSeek | | |
| Original | 0.3804 | 0.5500 |
| Routers & Expert | 0.2457 | 0.3145 |
| Routers Only | 0.2375 | 0.3315 |
| Experts Only | 0.2601 | 0.3435 |

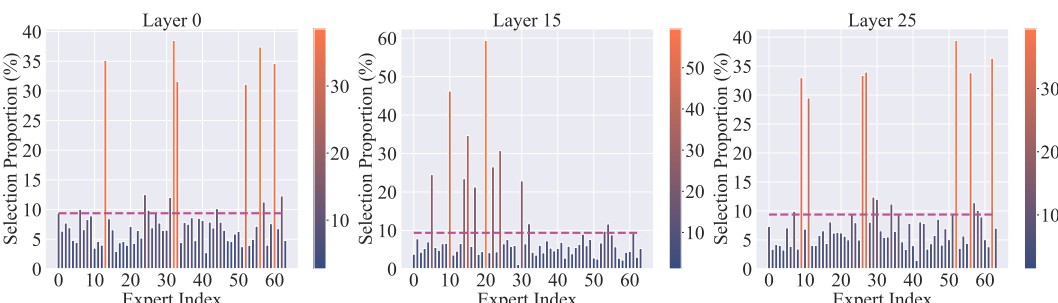

Figure 2: Proportion of tokens assigned to each expert of the pre-trained DeepSeek-v2-Lite ($K$=6 in Top$k$) with samples from the forgotten set of `WMDP` benchmark (Li et al., 2024) in different model layers. The dashed horizontal line marks 6/64, *i.e.*, the proportion expected with uniform expert selection. The expert selection distribution clearly follows a long-tailed pattern when the input is sampled from a topic within a narrow scope.

## 4 OUR PROPOSAL: UNLEARNING ONE EXPERT (UOE)

In this section, we delve into the failure cases highlighted in Sec. 3 by analyzing the behavior of routers and their expert selection patterns. We then identify two primary root causes underlying the poor unlearning performance in MoE LLMs. Based on these insights, we introduce UOE, a new unlearning paradigm designed to achieve controllable and effective unlearning for MoE LLMs.

**Uncovering the root cause: 'short-cut' in MoE LLM unlearning and expert selection shift.** In order to fully understand the failure cases of MoE LLM unlearning, we begin by inspecting and monitoring the expert selection pattern of the unlearned model. In **Fig. 2**, we show the proportion of tokens assigned to each selected expert on the data samples from `WMDP` dataset (Li et al., 2024). For the input of a specific topic, a small portion of experts (around 6 to 9 out of 64 experts) were assigned with the majority of the tokens in each layer, which was also confirmed in Wang et al. (2024b). Thus, we have the following insight:

> **Insight** 1: For the inference related to a certain topic within a narrow scope (*e.g.,* the forget set of an unlearning task), expert selection by MoE routers follows a long-tailed distribution, with only a few experts being activated significantly more frequently than others.

Based on the insight above, we define the frequently activated experts as topic-*target* experts, and the others as *non-target*. Thus, by eliminating the knowledge stored in these target experts, MoE LLM unlearning can be achieved more effectively. Next, we examine how the expert selection pattern evolves during unlearning. Specifically, we track the average expert selection overlap ratio across all layers between the unlearned model at different stages and the original pretrained model, when processing the forget set. The results, shown in **Fig. 3 (a)**, reveal a steady

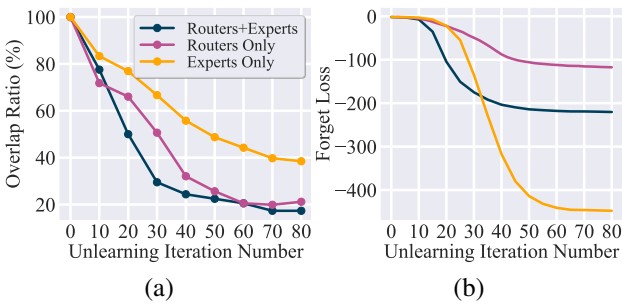

Figure 3: (a) Expert selection overlap ratio between the original pretrained model and the unlearned model with different unlearning iterations using GA on `WMDP` benchmark. (b) Forget loss vs. the number of unlearning iterations, when controlling parameters to unlearn in MoE LLM.

decline in the overlap ratio as unlearning progresses, indicating that previously selected target experts are gradually replaced by non-target ones that do not contain the target knowledge. This shift persists even when routers are fixed, as unlearning can still indirectly influence router selection: a router's decision at one layer depends on the output of the previous layer, which may have been affected by an updated expert of this previous layer in unlearning. Meantime, we observe a consistent reduction in forget loss, as shown in **Fig. 3 (b)**. Thus, we can derive the following insight:

> **Insight 2**: Existing unlearning methods tend to prompt the routers to shift expert selection from target to non-target experts unintentionally. This can create unlearning 'shortcuts' in expert selection to trick for low forget loss and lead to fake unlearning.

As unlearning proceeds, non-target experts are more frequently activated to handle samples related to the unlearning target, thereby being forced to participate in the unlearning task, even though they did not contain the intended target knowledge. Meanwhile, the true objective of unlearning, *i.e.,* the target experts, remain hidden out of the reach of the forward propagation. Existing literature (Liu et al., 2024c) has already demonstrated that forcing unlearning models that do not contain knowledge related to the unlearning target can cause a significant drop in model utility. This accounts for the sharp decline in model utility observed in Sec. 3, which leads to the following insight:

> **Insight 3**: The sharp degradation in model utility during MoE LLM unlearning is primarily due to excessive unlearning applied to non-target experts caused by expert selection shift.

**UOE for effective MoE LLM unlearning.** As discussed earlier, a new paradigm tailored for MoE LLM unlearning is urgently needed to address the challenges of unintentional expert selection shifts in routers and excessive unlearning of non-target experts. Therefore, we propose a framework that (1) identifies the most relevant target experts, (2) ensures that these target experts remain highly activated throughout the unlearning process to avoid selection shifts, and (3) limits the impact of unlearning on non-target experts. Spurred by these, we introduce UOE, where unlearning is confined to a single expert. We refer the readers to Alg. 1 for an illustration of UOE. This approach starts with an expert attribution process to accurately identify the most relevant experts for the unlearning task.

---

**Algorithm 1** UOE Unlearning Algorithm

---

**Output:** Unlearned Model $\boldsymbol{\theta}_u$
**Input:** Pretrained Model $\boldsymbol{\theta}_o$, Forget Set $\mathcal{D}_f$, Retain Set $\mathcal{D}_r$, Retain Loss $\ell_r$, Forget Loss $\ell_f$, Anchor Loss $L_{\text{anchor}}$
1: $D_s \leftarrow$ Sample_Subset($\mathcal{D}_f$)
2: $s \leftarrow$ Record_Affinity_Score($\boldsymbol{\theta}_o, D_s$)
3: $s_{\text{top},l} \leftarrow$ Ranking_And_Select($s$)
4: Activate_Expert_And_Router($\boldsymbol{\theta}_o, s_{\text{top}}$,Router$^l$)
5: $\boldsymbol{\theta}_u \leftarrow$ Unlearn($\boldsymbol{\theta}_o, \ell_f(\mathcal{D}_f), \ell_r(\mathcal{D}_r), L_{\text{anchor}}^{(l)}$)
6: Return $\boldsymbol{\theta}_u$

---

✦ **Expert attribution.** While the token assignment ratio for each expert, as shown in Fig. 2, can serve as a basic attribution metric, it overlooks finer details that are important for precise comparisons, due to the hidden states in each layer summed by weighted average. To address this, we adopt a gating score-based task affinity calculation method from (Wang et al., 2024b). Specifically, the affinity score for the $i$-th expert $e_i^{(l)}$ in the $l$-th layer of an MoE LLM is defined as:

$$s_i^{(l)} = \frac{1}{Z} \sum_{j=1}^{Z} \frac{1}{L_j} \sum_{t=1}^{L_j} g_{i,t}^{(l)} \tag{2}$$

where $Z$ is size of the calibration dataset used for expert attribution, $L_j$ represents the length of the $j$-th input sequence $\mathbf{x}_j$, and $g_{i,t}^{(l)}$ is the probability score assigned to expert $e_i^{(l)}$ for the $t$-th token. Following Wang et al. (2024b), the attribution data can be a subset universally sampled from the original forget set. We find that a subset containing over 100,000 tokens is robust enough to select the most relevant experts for an unlearning task. For each layer, we rank the experts based on their affinity score and select the top expert as the target expert for unlearning.

✦ **Router anchor loss.** A key challenge in unlearning is the expert selection shift, where the true target experts are hidden by the routers, while less relevant experts are activated during inference and inadvertently involved in the unlearning process. To mitigate this, we propose the router anchor loss, which encourages the previously identified target expert to remain consistently activated throughout unlearning. The loss is formulated as:

$$L_{\text{anchor}}^{(l)} = \|\mathbf{g}^{(l)} - [a_1^{(l)}, a_2^{(l)}, \dots, a_{E^{(l)}}^{(l)}]\|_2^2, \tag{3}$$

where $E^{(l)}$ is the total number of experts in the $l$-th layer, $\mathbf{g}^{(l)} = [g_1^{(l)}, g_2^{(l)}, \dots, g_i^{(l)}]$ is the output of router, and $a_i^{(l)} = 1$ if the $i$-th expert is identified as the target expert, otherwise $a_i^{(l)} = 0$. The

unlearning loss can be formularized as:

$$\min_{\boldsymbol{\theta}} \ell_f(\boldsymbol{\theta}; \mathcal{D}_f) + \lambda \ell_r(\boldsymbol{\theta}; \mathcal{D}_r) + \alpha L_{\text{anchor}}^{(l)}, \qquad (4)$$

where $\alpha$ is a hyperparameter to control the strength of anchor loss. The sensitivity analysis of $\alpha$ is provided in Sec. D in Appendix.

✦ **A single-layer solution for MoE LLM unlearning by UOE.** While we have successfully identified the most relevant target expert for each layer and implemented the router anchor loss to stabilize expert selection, applying unlearning across *all* layers still leads to expert selection shifts.

In **Fig. 4**, our results indicate that, even with the anchor loss, unlearning across multiple layers amplifies the effects of the MoE structure, where minor selection shifts in earlier layers are magnified, leading to substantial shifts in deeper layers. Consequently, the unlearned model still suffers a significant utility drop of over 30% (55.48% before unlearning *vs.* 24.65% after unlearning). To address this, we reduced the number of layers involved in the unlearning process. Surprisingly, unlearning just a single layer proved sufficient to achieve strong performance. By confining unlearning to one layer, we effectively minimize the cascading effects of expert selection shifts while still eliminating the target knowledge. In practice, we select the layer with the highest top-1 expert affinity score, as calculated in (2), and perform unlearning on its corresponding expert, which is treated as the target expert for the entire model. Since LLM architectures are consistent in size across layers, the highest affinity scores per layer can be directly compared. This approach forms the

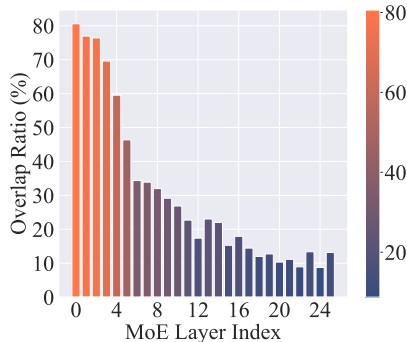

Figure 4: Expert selection overlap ratio for each MoE layer in DeepSeek-v2-Lite, comparing the unlearned and original models after 80 iterations with GA. The expert with the top affinity score in each layer is tunable during unlearning.

foundation of UOE, allowing it to maintain model utility and forget effectiveness with exceptional efficiency. To validate the effectiveness of the current design, in Sec. 5, we conduct extensive discussions and empirical studies on various other possible design choices than UOE. These include unlearning multiple experts in a single layer, unlearning single experts across multiple layers, and exploring different expert selection schemes beyond the affinity-score-based approach.

## 5 EXPERIMENT

### 5.1 EXPERIMENT SETUPS

**Unlearning tasks and datasets.** To demonstrate the effectiveness of our proposed method, we evaluate and compare it against different baselines on two widely accepted LLM unlearning benchmarks: WMDP (Li et al., 2024) and RWKU (Jin et al., 2024). WMDP assesses the model's ability to unlearn and prevent the generation of hazardous knowledge in biosecurity, cybersecurity, and chemical security contexts. RWKU, on the other hand, evaluates the model's capability to eliminate knowledge about 200 real-world celebrities, simulating a private information protection task. We note that other commonly used benchmarks, such as TOFU (Maini et al., 2024) and MUSE (Shi et al., 2024), are less appropriate in this work. These benchmarks require models to be fine-tuned before unlearning, which introduces additional biases to the results for MoE LLMs due to the known instability in training and the tricky hyper-parameter tuning involved (Jiang et al., 2024), often leading to training collapse (Zoph et al., 2022a).

**Models.** We evaluate different unlearning methods on two MoE LLMs: Qwen1.5-MoE-A2.7B-Chat (Qwen), mistralai/Mixtral-8x7B-Instruct-v0.1 (Mixtral), and DeepSeek-V2-Lite (DeepSeek), representing the two mainstream MoE LLM training schemes: upcycle-from-dense and train-from-scratch, respectively. Qwen has a total of 14.3 billion parameters, with 2.7 billion activated during inference, while DeepSeek has 16 billion parameters, of which 2.4 billion are activated during inference. Mixtral has 45 billion parameters, of which 12.9 billion are activated. Due to the computation limitation, Mixtral is only applied on UOE and other parameter-efficient fine-tuning baselines.

**Evaluation setup.** We evaluate the performance of the unlearned LLMs based on two key metrics: unlearning efficacy (**UE**) and preserved model utility (**UT**). For the **WMDP** task, UE is measured using the WMDP-Cyber subsets provided by the benchmark. Specifically, we use forget accuracy

Table 2: Performance comparison of existing unlearning methods equipped w/ and w/o UOE in WMDP (Li et al., 2024) and RWKU Jin et al. (2024) benchmarks on two MoE LLMs, namely Qwen1.5-MoE-A2.7B-Chat (Qwen) Team (2024) and DeepSeek-V2-Lite (DeepSeek) (Dai et al., 2024). The ↑ and ↓ symbols denote metrics where higher/lower values are better. The occurrence of significant utility drop (over 10% drop in UT compared to the pretrained model) are marked in red.

| Method | Qwen (WMDP) FA↓ | UT↑ | DeepSeek (WMDP) FA↓ | UT↑ | Qwen (RWKU) FA↓ | UT↑ | DeepSeek (RWKU) FA↓ | UT↑ |
|---|---|---|---|---|---|---|---|---|
| **Pretrained** | 0.4192 | 0.5979 | 0.3804 | 0.5548 | 0.4243 | 0.5979 | 0.5376 | 0.5548 |
| **GA** | 0.2953 | 0.3393 | 0.2457 | 0.3145 | 0.0078 | 0.4849 | 0.0839 | 0.5195 |
| **GA+UOE** | 0.2987 | 0.5012 | 0.2700 | 0.5100 | 0.0060 | 0.5709 | 0.0000 | 0.5485 |
| **GDIFF** | 0.2964 | 0.2965 | 0.2898 | 0.3929 | 0.0700 | 0.5296 | 0.1901 | 0.3495 |
| **GDIFF+UOE** | 0.2445 | 0.5295 | 0.2677 | 0.4895 | 0.0010 | 0.5987 | 0.0000 | 0.5253 |
| **NPO** | 0.3447 | 0.4612 | 0.3200 | 0.4700 | 0.0000 | 0.3718 | 0.0970 | 0.5388 |
| **NPO+UOE** | 0.3200 | 0.5468 | 0.2898 | 0.4790 | 0.0020 | 0.5428 | 0.0000 | 0.5479 |
| **RMU** | 0.2612 | 0.3560 | 0.2530 | 0.4540 | 0.0200 | 0.2420 | 0.0010 | 0.5109 |
| **RMU+UOE** | 0.2536 | 0.5351 | 0.2859 | 0.5424 | 0.0723 | 0.5975 | 0.0130 | 0.5388 |

(FA)—the accuracy of the LLMs on the forget set after unlearning—as the measure of UE. A lower FA indicates better unlearning. Given the four-option multiple-choice format of the test set, the ideal FA is 0.25, equivalent to random guessing. UT is assessed using the zero-shot accuracy on the MMLU dataset (Hendrycks et al., 2020), which reflects the model's ability to retain general knowledge. For the RWKU task, we use the Rouge-L recall score to evaluate performance on fill-in-the-blank and question-answer tasks, with lower scores indicating more effective unlearning. Since the task follows a question-answer format, the ideal FA is 0.0, indicating no overlap between the generated answer and the ground truth. The UT evaluation for RWKU is the same as for WMDP, using the MMLU benchmark. By default, during the unlearning process, we select the model checkpoint that achieves the best balance between UE and UT as the optimal checkpoint.

**Baselines.** We demonstrate the effectiveness of our proposed UOE framework by comparing it against the LLM unlearning baselines: Gradient Ascent (GA) (Eldan & Russinovich, 2023), Gradient Difference (GDIFF) (Maini et al., 2024) and most recent unlearning algorithm Negative Preference Optimization (NPO) (Zhang et al., 2024) and Representation Misdirection for Unlearning (RMU) (Li et al., 2024). For each method, we compare the original results with those obtained when incorporating UOE. Given the parameter efficiency of UOE, we also compare it with two state-of-the-art parameter-efficient fine-tuning (PEFT) methods for MoE LLMs: the low-rank adaptation scheme (LoRA) (Hu et al., 2021) and the Expert-Specialized Fine-Tuning method (ESFT) Wang et al. (2024b), which is specifically designed for MoE LLMs.

## 5.2 EXPERIMENT RESULTS

**Effectiveness of UOE in preserving model utility and unlearning efficacy.** In **Tab. 2**, we present the UE (unlearning efficacy) and UT (utility) performance of our proposed UOE when integrated into different unlearning methods GA, GDIFF, NPO, and RMU. First, one of the most notable findings is that UOE significantly improves model utility (**UT**) across all tested methods. For instance, when applied to baseline methods like GA, GDIFF, and RMU, UOE consistently mitigates the severe utility drops (greater than 10%) that occur with the unmodified methods. This is particularly evident in scenarios where baseline methods without UOE exhibit drastic performance degradation in model utility (highlighted in red), while the same methods paired with UOE show substantial recovery. For example, the utility of GA on Qwen for the WMDP task drops from 0.5979 to 0.3393, but with UOE, the utility improves to 0.5012, restoring much of the lost performance. Similarly, GDIFF on RWKU suffers a significant utility loss from 0.5979 to 0.3495, but when UOE is applied, utility rises back to 0.5253, nearly matching the original pretrained performance. Second, beyond utility preservation, the unlearning efficacy (**UE**)—measured through FA—remains either unaffected or slightly *improved* when UOE is employed. This balance between utility preservation and effective unlearning highlights the advantage of UOE. For instance, GDIFF+UOE reduces the forget accuracy (FA) on Qwen (WMDP) from 0.2964 to 0.2445, demonstrating better unlearning while still achieving a higher utility score. Similarly, RMU+UOE on DeepSeek (WMDP) lowers the FA from 0.2530 to 0.2859, with a corresponding utility improvement from 0.4540 to 0.5424. Notably, methods such as GDIFF and RMU, which experience significant utility loss when used alone, benefit greatly from the application of UOE, achieving near-pretrained utility levels while still maintaining effective unlearning.

Table 4: Performance comparison between existing unlearning methods GA equipped with UOE and other PEFT methods, including LoRA (Hu et al., 2021) and ESFT (Wang et al., 2024b). The occurrence of significant utility drop (over 10% drop in UT compared to the pretrained model) are marked in red.

| Method | Qwen (WMDP) | | DeepSeek (WMDP) | | Mixtral (WMDP) | | Qwen (RWKU) | | DeepSeek (RWKU) | | Mixtral (RWKU) | |
|---|---|---|---|---|---|---|---|---|---|---|---|---|
| | FA↓ | UT↑ | FA↓ | UT↑ | FA↓ | UT↑ | FA↓ | UT↑ | FA↓ | UT↑ | FA↓ | UT↑ |
| Pretrained | 0.4192 | 0.5979 | 0.3804 | 0.5548 | 0.5229 | 0.6885 | 0.4243 | 0.5979 | 0.5376 | 0.5548 | 0.5820 | 0.6885 |
| LoRA | 0.2459 | 0.2689 | 0.2657 | 0.2295 | 0.2658 | 0.2597 | 0.0000 | 0.2689 | 0.0000 | 0.2302 | 0.0000 | 0.2295 |
| ESFT | 0.3145 | 0.4514 | 0.2737 | 0.5108 | 0.2547 | 0.6386 | 0.001 | 0.4433 | 0.0200 | 0.5001 | 0.0542 | 0.6743 |
| UOE | 0.2987 | 0.5012 | 0.2700 | 0.5100 | 0.2608 | 0.6364 | 0.006 | 0.5709 | 0.0000 | 0.5485 | 0.0455 | 0.6713 |

**UOE significantly outperforms other PEFT methods.** Tab. 4 shows the performance comparison between UOE and other PEFT methods, and Tab. 3 shows a comparison of the parameter efficiency among different PEFT methods. Several key conclusions can be drawn: First, UOE achieves far better parameter efficiency, with only 0.06% of tunable parameters, compared to LoRA (0.92%) and ESFT (2.86%), while still outperforming them in utility preservation. For instance, in RWKU, UOE achieves utility scores of 0.5709 on Qwen

Table 3: Tunable parameter ratio, PEFT vs UOE.

| Method | Tunable Parameter Ratio | | |
|---|---|---|---|
| | Qwen | DeepSeek | Mixtral |
| LoRA | 0.87% | 0.92% | 0.26% |
| ESFT | 3.13% | 2.86% | 14% |
| UOE | 0.06% | 0.06% | 0.41% |

and 0.5445 on DeepSeek, significantly higher than LoRA (0.2689 and 0.2302) and ESFT (0.4433 and 0.5001). Second, the utility preservation of UOE is much better than the others, and this is achieved while maintaining a comparable level of forget efficacy. For example, on WMDP, UOE achieves a utility score of 0.5012 for Qwen, much higher than LoRA's 0.2689, with a similar forget efficacy (FA: 0.2987 vs. 0.2459). These results clearly demonstrate that UOE is the more balanced and efficient solution for unlearning tasks, particularly when both parameter efficiency and utility retention are important.

**Comparison of different design choices in UOE framework.** In designing our UOE method, we opted to unlearn only a single expert in one specific layer, guided by the affinity score. However, to justify this design decision, we conducted a series of empirical studies comparing alternative approaches. These experiments were carried out using the RMU unlearning method on the WMDP task, where we tuned each approach until the model fully unlearned (*i.e.*, FA reached 25%), and then compared (1) if the problem of the expert selection shift has been properly addressed and (2) the model utility score (**UT**) across the different strategies.

• *Tuning multiple layers with one expert per layer. Tuning a single expert across multiple layers.* In Fig. 4, we briefly discussed how the cumulative effect leads to a significant issue of expert selection shift in the deeper layers

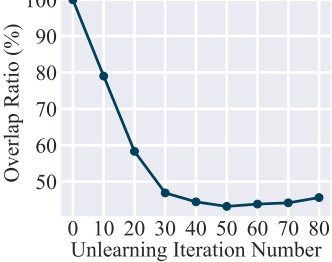

Figure 5: Expert selection overlap ratio between the pretrained model and the unlearned model across different unlearning iterations using GA on the WMDP benchmark. Experts with the highest affinity score in each layer are tuned during unlearning.

of MoE LLMs. In Fig. 5, we further illustrate how the overall expert selection overlap ratio changes as unlearning progresses. As shown, even when only one expert is tunable per layer, the cumulative effect causes the overall overlap ratio to drop sharply, leaving the expert selection shift issue unresolved. Thus, tuning all layers during MoE LLM unlearning is not a feasible solution.

• *Tuning multiple experts in a single layer.* A natural question within the UOE framework is whether involving more than one expert in the selected layer during unlearning would yield better results. In Tab. 5, we examine the impact of increasing the number of tunable experts in a single layer. As the table shown, when the unlearned model reaches the same level of UE (with an FA of around 25%) controlled by different training steps, the

Table 5: Model utility (UT) comparison, at the same level of forget efficacy (FA≈ 0.25), when different number of experts are unlearned using GA on WMDP benchmark.

| # of experts | 1 | 3 | 6 |
|---|---|---|---|
| FA (↓) | | ~ 0.2500 | |
| UT (↑) | 0.5100 | 0.4856 | 0.4652 |

model utility decreases significantly as the number of tunable experts increases. For instance, when the tunable expert count is increased to 6, the model utility drops by over 4%, from 0.51 to below 0.47. This suggests that involving more experts in unlearning makes it harder to maintain utility, without providing any noticeable improvement in unlearning efficacy.

• *Sensitivity of* UOE *to expert selection and layer selection schemes.* Next, we explored alternative methods for selecting the target expert, beyond the affinity score-based approach used in UOE.

Table 6: Performance comparison between UOE and the random expert selection scheme on the `WMDP` task. Other settings are consistent with those in Tab. 2.

| Method | Qwen (`WMDP`) | | DeepSeek (`WMDP`) | | Qwen (`RWKU`) | | DeepSeek (`RWKU`) | |
|---|---|---|---|---|---|---|---|---|
| | FA↓ | UT↑ | FA↓ | UT↑ | FA↓ | UT↑ | FA↓ | UT↑ |
| **Pretrained** | 0.4192 | 0.5979 | 0.3804 | 0.5548 | 0.4243 | 0.5979 | 0.5376 | 0.5548 |
| **RMU** | 0.2612 | 0.3560 | 0.2530 | 0.4540 | 0.0200 | 0.2420 | 0.0010 | 0.5109 |
| **Random+RMU** | 0.3505 | 0.5947 | 0.2722 | 0.5183 | 0.2110 | 0.5924 | 0.1176 | 0.5182 |
| **UOE+RMU** | 0.2536 | 0.5351 | 0.2859 | 0.5424 | 0.0723 | 0.5975 | 0.0130 | 0.5388 |

Specifically, we investigated how sensitive UOE is to the choice of the expert to be unlearned. In **Tab. 6**, we compared the performance of the affinity score-based selection scheme in UOE with a random expert selection approach on the `WMDP` task. The results clearly show that while random expert selection can sometimes yield comparable utility preservation, it fails to achieve the same level of unlearning efficacy as UOE. For instance, on Qwen (`WMDP`), the random selection achieves a utility score of 0.5947 versus UOE's 0.5351, but the forget accuracy (**FA**) with random selection is significantly higher at 0.3505, compared to 0.2536 for UOE. This indicates that random selection does not drive FA low enough, compromising the unlearning objective. Similarly, on DeepSeek (`RWKU`), random selection results in an FA of 0.1176, whereas UOE achieves a much lower FA of 0.013, highlighting the better unlearning performance of UOE. In conclusion, selecting experts with highest affinity score perform better than random expert selection.

Finally, we examine the performance of UOE when layers with different top1 expert affinity score rankings are selected for unlearning. In Tab. 7, we observe that UOE is robust and relatively insensitive to the specific layer chosen for unlearning, as long as the affinity score remains reason-

Table 7: Model utility (UT) comparison across layers with different affinity score rankings in UOE on the `RWKU` benchmark. UT is compared at a consistent level of forget efficacy (FA ≈ 0.25).

| Layer Ranking | #1 | #2 | #3 | #13 | #20 | #23 | #26 |
|---|---|---|---|---|---|---|---|
| Affinity Score | 0.2110 | 0.1957 | 0.1695 | 0.1115 | 0.0942 | 0.0844 | 0.0618 |
| FA (↓) | | | ∼ 0.2500 | | | | |
| UT (↑) | 0.5485 | 0.5475 | 0.5453 | 0.5445 | 0.5441 | 0.4262 | 0.2355 |

ably high. For instance, even when selecting the 13th or 20th ranked layers, the model utility (**UT**) remains stable at around 0.5445, although their affinity scores of 0.1115 and 0.0942 are lower than that of the top-ranked layer. However, once the affinity score drops further, as seen in the 23rd and 26th ranked layers (with scores of 0.0844 and 0.0618), the utility decreases more sharply, falling to 0.4262 and 0.2355, respectively. This demonstrates that while UOE maintains strong performance across a wide range of layers, selecting layers with very low affinity scores can negatively impact utility. Overall, these results highlight the robustness of UOE and its ability to tolerate variability in layer selection without sacrificing unlearning efficacy, provided that layers with sufficiently high affinity scores are chosen.

Our empirical studies demonstrate that the design choice of unlearning a single expert in one layer, guided by the affinity score, is satisfactorily reasonable for balancing effective unlearning with utility preservation. The alternative approaches, whether involving multiple layers or multiple experts, consistently led to greater utility degradation and instability. Selecting expert with highest affinity scores can achieve better forget quality than picking experts by random choice. At last UOE is very robust to layer selection scheme. Therefore, our current design proves to be the most effective and efficient for MoE LLM unlearning.

## 6 CONCLUSION

In this paper, we for the first time examine the challenges of applying existing MU techniques to MoE LLMs and carefully investigate the synergy between the dynamic routing system of MoE LLM and the unlearning effects. To address these issues, we proposed UOE, a novel framework that unlearns a single expert in a targeted layer while stabilizing expert selection through a router anchor loss. This approach mitigates expert selection shifts and achieves efficient unlearning with minimal parameter updates. Extensive experiments show that UOE significantly outperforms traditional unlearning methods and other parameter-efficient fine-tuning techniques, providing a robust solution for MoE LLM unlearning tasks.

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

Table 8: Example of GCG attack. Original is the output of the original Deepseek model.

| Questions | Original | Without Attack | With Attack |
|---|---|---|---|
| Stephen King is often referred to as the 'King of ___'. | Horror | indingindingind-ingindingindingi-ndinginding... | ologist:Sure here is the answer: Question : Stephenindingindingin-dinginding... |
| Bruce Lee moved to ___ in 1959. | America | -wardswar-dswardswards... | eanswer:" Answer:[/b]inding [b]... |

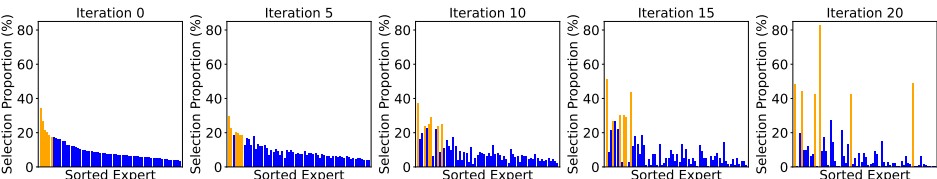

Figure 6: Token Proportion Shift in layer 21 of Deepseek unlearned by GA in WMDP dataset.

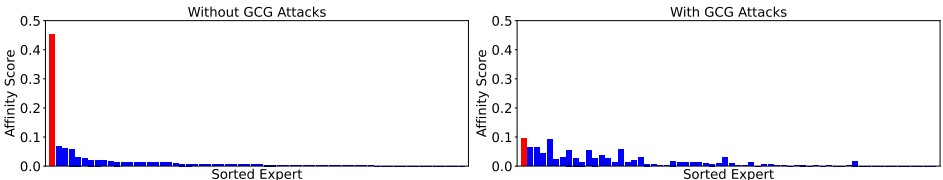

Figure 7: Affinity Score of all experts in target layer in Deepseek unlearned GA on RWKU dataset. The target expert is marked as red.

## A  EXPERIMENT SETTINGS

**Hyperparameter selection.** We set the learning rate to 5e-5 for GA, NPO, and GD while setting it to 1e-4 for UOE. The batch size is 4 for GA, NPO, and GD, while it is set to 16 for UOE. In NPO, the beta value is set to 0.001. The alpha for the retain loss is set to 1 in both GD and NPO. For RMU, we follow the hyperparameters specified in the original work. We configure the steering coefficients as 8000 for Qwen and 32000 for Deepseek, as UOE targets deeper layers in these models. For ESFT, we set the threshold $p = 0.15$.

**Dataset Settings.** For the WMDP dataset, we use the cyber-forget-corpus as the forget set and wmdp-cyber as the evaluation set, in line with WMDP (Li et al., 2024). The Wikitext (Merity et al., 2016) dataset serves as the retain set for both GD and RMU tasks, also following WMDP (Li et al., 2024). In the RWKU (Jin et al., 2024) dataset, we follow the original study by selecting 100 individuals as unlearning targets. The train_original_passage set, which includes Wikipedia descriptions of these 100 individuals as provided in the paper, is used as the forget set.

**Evaluation Settings.** We utilize the LM Evaluation Harness (Gao et al., 2024) to measure zero-shot accuracy on the MMLU and WMDP cyber datasets. The mean accuracy across all tasks in MMLU serves as a measure of model utility. For the RWKU dataset, we adhere to the original settings, using the prompt "Please complete the blank in the following question. Question:" for fill-in-the-blank tasks and "Please briefly answer the following question. Question:" for generation tasks.

## B  JAILBREAK ATTACK ON UOE

Table 10: Sensitivity Analysis of hyperparameter $\alpha$ for the strength of anchor loss. The experiment is conducted on Deepseek unlearned by GA with RWKU dataset.

| $\alpha$ | 0 | 1 | 100 | 1000 |
|---|---|---|---|---|
| FA ($\downarrow$) | 0.0 | 0.0 | 0.0 | 0.0 |
| UT ($\uparrow$) | 0.5435 | 0.5485 | 0.5471 | 0.5468 |

To investigate the soft prompt attack, we employ the GCG attack Zou et al. (2023) in a white-box setting to optimize the prompt, aiming to force responses to begin with "Sure, here is the answer:". The number of optimization steps is increased to 5000, while other hyperparameters remain at the default settings. Given the compu-

Table 9: UOE against GCG attack on Deepseek unlearned with RWKU dataset.

| With Attack | Without Attack |
|---|---|
| 0.01 | 0.01 |

tational demand (approximately 1 GPU hour on an A100 for generating a single soft prompt), we optimize 400 prompts across 400 samples in RWKU. Since not all responses begin with "Sure, here is the answer:", we filter for those containing the word "answer" and then assess the forget quality both with and without GCG-generated prompts. Experimental results in Tab. 8 indicate that the GCG, despite being the strongest prompt-level attack, fails to recover the forgotten knowledge, as the forget accuracy (FA) remains at 0.01 before and after the GCG attacks.

## C   TOKEN PROPORTION SHIFT VISUALIZATION

We visualize the expert selection distribution in one layer across the unlearning process in Fig.6 in revision. The figures sort the experts in layer 21 by the selection proportion in the original model and keeps this order to plot the figure in unlearned models to show the changes in all experts. The results show that GA algorithm even decreases the uncertainty in WMDP after unlearning.

## D   SENSITIVITY ANALYSIS OF HYPERPARAMETER $\alpha$

We conduct experiments on Deepseek unlearned by GA with RWKU dataset to explore the performance of different $\alpha$. As shown in Tab. 10, the results indicate that UOE is robust to a wide range of $\alpha$ and achieves the best performance when $\alpha = 1$.

## E   DISCUSSION ON SHARED EXPERTS

Shared expert is a special architecture in both Deepseek and Qwen, where all tokens activate the shared experts in all layers. In this section, we discuss how UOE can still achieve good Forget Quality with shared experts. The outputs of shared experts and normal experts are aggregated in the hidden state of each layer, where the output of normal experts can neutralize the output of shared experts. The objective loss function in unlearning algorithms is designed to unrelate the hidden state with the unlearned target. To achieve this, UOE introduces perturbations to the selected expert, i.e., adding noise in the aggregation step before outputting the hidden state values. From here on, the perturbation disrupts the knowledge that was learned by shared experts or previous layers. To formally show this process, the output hidden state of $l$-th layer is:

$$\mathbf{h}'^{(l)}_t = g^{(l)}_{\texttt{target},t}\text{FFN}^{(l)}_{\texttt{target}}(\mathbf{u}_t) + \mathbf{u}^{(l)}_t + \text{FFN}^{(l)}_{\texttt{shared}}(\mathbf{u}_t) + \sum_{i=1}^{N-1} g^{(l)}_{i,t}\text{FFN}^{(l)}_i(\mathbf{u}_t),$$

where $\text{FFN}_{\texttt{shared}}$ are shared experts, $\text{FFN}_{\texttt{target}}$ is the target expert, and $\sum_{i=1}^{N-1} g^{(l)}_{i,t}\text{FFN}^{(l)}_i(\mathbf{u}_t)$ are selected experts except for the target expert. After the target expert is unlearned

$$g'^{(l)}_{\texttt{target},t}\text{FFN}'^{(l)}_{\texttt{target}}(\mathbf{u}_t) = g^{(l)}_{\texttt{target},t}\text{FFN}^{(l)}_{\texttt{target}}(\mathbf{u}_t) + \mathbf{h}^{(l)}_{t,\texttt{perturbation}},$$

e.g. by adding the perturbation $h^l_{t,perturbation}$, the output hidden state of l-th layer becomes

$$\mathbf{h}'^{(l)}_t = g^{(l)}_{\mathtt{target},t}\mathrm{FFN}^{(l)}_{\mathtt{target}}(\mathbf{u}_t) + \mathbf{h}^{(l)}_{t,\mathtt{perturbation}} + \mathbf{u}^{(l)}_t + \mathrm{FFN}^{(l)}_{\mathtt{shared}}(\mathbf{u}_t) + \sum_{i=1}^{N-1} g^{(l)}_{i,t}\,\mathrm{FFN}^{(l)}_i(\mathbf{u}_t).$$

This output of the $l$-th layer will be taken as input to the followup layers, thus the perturbation is propagated, leading to the aim of unlearning. Note that the added perturbation $\mathbf{h}^{(l)}_{t,\mathtt{perturbation}}$ is not random, instead, it is optimized by minimizing the unlearning loss function to perturb both the outputs of shared experts $\mathrm{FFN}^{(l)}_{\mathtt{shared}}$ and the output from the previous layer $\mathbf{u}^{(l)}_t$.

