# OpenReview forum: "UOE: Unlearning One Expert is Enough for Mixture-of-Experts LLMs"
_ICLR.cc/2025/Conference — Submitted to ICLR 2025_

### Official Review · Reviewer_JQDp · 2024-10-28

**Soundness:** 2
**Presentation:** 3
**Contribution:** 2
**Rating:** 5
**Confidence:** 4

**Summary:**

This paper investigated a critical problem when applying machine unlearning in an MoE scenario. The author gives an insightful demonstration of the current challenge (e.g., Fig 1(a) and Fig 2), making this paper easy to follow. As the authors mentioned, the short-cut behavior when the MoE system is doing expert selection leads to unstable unlearning. To address this issue, the authors proposed an Unlearning One Expert system, which prevents the frequent switching of expert selection and ensures a focused, controlled, and stable unlearning. Although most of this paper is written clearly, I still have concerns about motivation, contribution, and technical quality.

**Strengths:**

S1. The research problem is well formalized and presented with some demonstration of current research limitation visualization. \
S2. The authors provided a detailed introduction to recent studies on MoE, MU, etc. \
S3. The methodology and most of the experiment parts are well-written.

**Weaknesses:**

W1. Regarding the motivation for applying MU tech to the MoE system. There could be a wild ground for application, but the technical challenge remains unclear. One key issue is raised due to the golden label of expert selection being decided during MoE system training. It could be more reasonable to directly apply this _decision + MU tech_ when unlearning on the MoE system remains unclear.

W2. Like some routing algorithms, the performance improvement result from whether to reduce the uncertainty or routing token to correct expert is unclear.  The author could provide more experimental detail to clarify the contribution of the UOE system.

W3. There are many routing algorithms proposed recently. However, UOE seems only to use TopK routing as the backbone. The versatility of UOE framework is not discussed well.

**Questions:**

Major Questions:

Q1. Is the rising **uncertainty** among routing algorithms causing ineffective MoE unlearning? If so, why not directly optimize (or maintain) the entropy?

Q2. As $\lambda$ (hyperparameter for retaining loss, eq.1) aims to directly optimize the strength of retain quality, there is no discussion on this setting. If $\lambda$ is set to $0$, what would happen?

Q3-1. Is the affinity score unchanged (or as a golden label) during the unlearning progress?

Q3-2. If the affinity score is the golden target, why not use it as a roadmap but an optimization target? Is it proper to refine the routing strategy during optimization?

Q4. Some case studies on generated unlearning data could be interesting. What will happen in the MoE system?

Minor Questions:

MQ1. In Fig. 3 (a-b), what is the difference between the settings for Routers+Experts, Routers Only, and Experts Only? Why do those settings matter?\
MQ2. Also, in Fig. 3 (a-b), will the deeper layers affect the routing overlap ratio?

---

> ### Author Response · Authors · 2024-11-21
>
> W1-1:  the technical challenge of unlearning MoE.
>
> A: Before our study, there has been no research on unlearning for MoE LLMs. A straightforward approach would be to apply existing MU techniques to them. However, as mentioned in Section 3, we find that this naive solution fails due to the expert selection shift issue. As discussed in Insight 2 in Section 4, this issue arises because the dynamic routing mechanism in MoE shifts expert selection from tar-get to non-target experts unintentionally. This unique challenge posed by the MoE architecture motivates our proposed approach UOE.
>
> W1-2: It could be more reasonable to directly apply this decision + MU tech when unlearning on the MoE system remains unclear.
>
> A: Thank you for the suggestion. We would like to discuss the decision + MU if you can provide more information about what is the “decision” here.
>
> W2: Like some routing algorithms, the performance improvement result from whether to reduce the uncertainty or routing token to the correct expert is unclear.
>
> Q1: Is the rising uncertainty among routing algorithms causing ineffective MoE unlearning? If so, why not directly optimize (or maintain) the entropy?
>
> A: Thanks for the valuable question. According to [1], uncertainty measures routing results of expert selection of tokens. We visualize the expert selection distribution in one layer across the unlearning process in Fig.6 in revision. The series of figures sort the experts in layer 21 by the selection proportion in the original Deepseek model and keep this order to plot the figures in unlearned models to show the changes in all experts. The results show that GA algorithm decreases the uncertainty in WMDP after unlearning. We would like to have further discussion with the reviewer regarding if this is an interesting analysis connecting the decrease of uncertainty to the effectiveness of unlearning by our UOE.
>
> > [1] Wu, Haoze, et al. "GW-MoE: Resolving Uncertainty in MoE Router with Global Workspace Theory." arXiv preprint arXiv:2406.12375 (2024).
>
> W3: UOE seems only to use TopK routing as the backbone. The versatility of UOE framework is not discussed well.
>
> A: We believe that our methods are versatile and can be applied to both sparse and dense routing schemes. In the case of dense routing, where all experts contribute to the final output, affinity scores can still be calculated to measure the relevance of each expert's contribution, similar to how they are utilized in top-K routing.
>
> Q2: As λ (hyperparameter for retaining loss, eq.1) aims to directly optimize the strength of retain quality, there is no discussion on this setting. If λ is set to 0, what would happen?
>
> A: If $\lambda$ is set to 0, GD will be the same as GA. In our experiment, both GA and NPO set $\lambda$ to 0. The results in Tab. 2 in our original submission indicate that GA+UOE shows better performance than GD+UOE.
>
> Q3-1: Is the affinity score unchanged (or as a golden label) during the unlearning progress?
>
> A: No, the affinity score does change, which leads to the shift in expert selection. As shown in Fig. 6 of our revision, the affinity scores change significantly during the GA unlearning process. In UOE+GA, we observe that the top-1 expert's affinity score increases from 0.21 to 0.53 after unlearning.
>
> Q3-2: If the affinity score is the golden target, why not use it as a roadmap but an optimization target? Is it proper to refine the routing strategy during optimization?
>
> A: In our work, we aim to avoid making changes to the architecture of MoE models. It may be a good idea to apply a roadmap, but there are some challenges. First, identifying target tokens and determining whether to unlearn within the same token is complex. For example, we may want to unlearn code related to cybersecurity issues but retain coding capabilities in MMLU. Second, it is unclear how to adapt MoE models with an optimization strategy to roadmap routing for all tokens in both the unlearn and retain sets. We plan to explore these issues in the future.

---

> > ### Author Response · Authors · 2024-11-21
> >
> > Q4: Some case studies on generated unlearning data could be interesting. What will happen in the MoE system?
> >
> > A: Thanks for the suggestion. We included in our revised draft some case studies of the unlearned model in Tab. 8, as well as how it performs against the CGC attack.
> >
> > | Questions                                               | Original Answer | Unlearn without soft prompt                 | Unlearn with soft prompt                                     |
> > | ------------------------------------------------------- | --------------- | ------------------------------------------- | ------------------------------------------------------------ |
> > | Stephen King is often referred to as the 'King of ___'. | Horror          | indingindingindingindingindingindinginding… | ologist:Sure here is the answer:Question : Stephenindingindingindinginding… |
> > | Bruce Lee moved to ___ in 1959.                         | America         | -wardswardswardswards...                    | eanswer:” An-swer:[/b]inding [b]...                          |
> >
> > MQ1:  In Fig. 3 (a-b), what is the difference between the settings for Routers+Experts, Routers Only, and Experts Only? Why do those settings matter?
> >
> > A: These settings were designed specifically to explore the difficulties and potential solutions for addressing the problem of unlearning in MoE LLMs. In the Routers+Experts setting, all routers and experts participate in unlearning, similar to the process used for unlearning in non-MoE dense models. The results of this setting demonstrate the failure of existing unlearning algorithms, which were designed for non-MoE models, when applied to the entire MoE architecture. In the Routers Only setting, only the router layers undergo unlearning. The results indicate that misattributing tokens to irrelevant experts negatively impacts utility performance. In the final setting, Experts Only, all expert layers are unlearned. The results reveal that unlearning MLP layers—considered the primary storage of knowledge—significantly harms the model's utility. Overall, the results across all settings highlight the unique challenges posed by the MoE architecture for unlearning. To address these challenges, we propose our flexible framework, UOE.
> >
> > MQ2: Also, in Fig. 3 (a-b), will the deeper layers affect the routing overlap ratio?
> >
> > A: Yes, as shown in Figure 4, deeper layers will be affected greater than shallow layers.

---

> ### Comment · Reviewer_JQDp · 2024-11-21
> **Further discuss**
>
> Regarding W1-2: Decision refers to the expert-selection (or the weight score for dense routing). For instance, we could know where one token is routing to.
>
> Regarding Q4: Very interesting observation, could you further analysis the underlying driver?

---

> > ### Author Response · Authors · 2024-11-22
> >
> > Q1: Regarding W1-2: Decision refers to the expert-selection (or the weight score for dense routing). For instance, we could know where one token is routing to.
> >
> > A: To map tokens to irrelevant experts, the Router Only experiment introduces a perturbation in expert selection, as shown in Table 1 of the Preliminaries section. The results demonstrate that misattributing tokens to irrelevant experts significantly degrades utility performance. Therefore, Decision+MU requires a refined design to achieve optimal performance in terms of Model Utility.
> >
> > Q2: Regarding Q4: Very interesting observation, could you further analysis the underlying driver?
> >
> > A: Thank you for the questions. The GCG attack operates in a white-box setting to optimize the prompt, aiming to elicit responses starting with “Sure, here is the answer:”. We visualize the affinity scores of the unlearned model with and without GCG attack prompts. The results (Fig. 7 in our revision or the [Anonymous link](https://imgur.com/a/zyiVam4)) show that, while the GCG attack reduces the affinity score for the target expert, the target expert remains the top-1 in affinity score and continues to dominate the weighted average.

---

> > > ### Author Response · Authors · 2024-11-27
> > > **Thanks for your insightful review**
> > >
> > > We would like to follow up to see if our additional contributions effectively addressed your comments. If they did, we kindly ask if you could update your score accordingly. If not, could you please provide more specific feedback on what we could improve or clarify during the rebuttal period? We would be happy to address any additional points.

---

> > > > ### Author Response · Authors · 2024-12-02
> > > > **Thanks for your valuable feedback**
> > > >
> > > > Thank you for taking the time to review our work and provide valuable feedback. If you have no further questions or concerns, we kindly ask if you could consider adjusting your score accordingly.

---

### Official Review · Reviewer_QbMP · 2024-11-02

**Soundness:** 2
**Presentation:** 2
**Contribution:** 3
**Rating:** 5
**Confidence:** 3

**Summary:**

This paper looks into unlearning (UL) for Mixture-of-Experts LLMs, being a pilot study in this area. It is found that existent UL algorithms suffer from expert selection shifts and thus cannot efficiently eliminate certain knowledge. This paper offers a simple solution that is to unlearn the target expert in a single layer. Experiments support the effectiveness of the method.

**Strengths:**

- This paper identifies the unique challenge of unlearning MoE LLMs.

- The experiments are thoroughly conducted and ablate well on the design components of UoE such as the number of layers/experts to unlearn and sensitivity to expert ranking.

**Weaknesses:**

- My primary concern is that both Qwen1.5 MoE [1] and DeepSeek MoE [2] have shared experts, meaning certain experts are always activated and are not subject to router choices. The paper does not mention how this issue is addressed. I believe these shared experts could retain the knowledge of the forget set. The paper lacks discussion on this point, and additional evidence is needed.

- The unlearning loss objective is unclear, is it a combination of (1) and (3)? What is the hyperparameter applied to the anchor loss, and how sensitive is model performance to this hyperparameter?


[1] https://qwenlm.github.io/blog/qwen-moe/

[2] https://huggingface.co/deepseek-ai/DeepSeek-V2-Lite

**Questions:**

- What does line 4 in Algorithm 1 mean? Only router parameters in the same selected layer are activated accordingly, or all router parameters?

- Related to what is pointed out in Weakness. The affinity score is only taken from router outputs and thus ignores shared experts. Is there a reason to ignore those shared experts?

- For other MoE models (for example Mixtral) without such a shared expert setting, how does UoE work?



I believe further clarification is needed on these points, and I am open to adjusting my scores if my concerns are addressed.

---

> ### Author Response · Authors · 2024-11-21
>
> W1: The discussion on shared experts during unlearning, and additional evidence is needed.
>
> A: Thank you for the suggestion. Shared experts are designed to capture and consolidate common knowledge across different contexts [2]. However, our task focuses on more specific knowledge, which is less likely to be represented by shared experts. Additionally, the top-1 expert after unlearning introduces perturbation to the knowledge that is propagated from previous layers and shared experts. This influence helps ensure that the targeted knowledge is effectively disrupted throughout the network. To explore the outcome of unlearning shared experts, we conduct experiments on Deepseek unlearned by GA with WMDP. The results below indicate that unlearning shared experts leads to a sharp decrease in utility, e.g., the Forget Accuracy drops to 0.3554 when 4 shared experts are unlearned along with the top-1 expert, in comparison, the Forget Accuracy is 0.51 when only unlearning the top-1 expert. Even unlearning the top-6 experts (6 experts as targets) is better than unlearning four shared experts.
>
> > [2] Liu, Aixin, et al. "Deepseek-v2: A strong, economical, and efficient mixture-of-experts language model." arXiv preprint arXiv:2405.04434 (2024).
>
> | Experts | Top 1  | Top 1+ 4 Shared | Top 6  |
> | ------- | ------ | -------------- | ------ |
> | FA ↓    | 0.2700 | 0.2657         | 0.2567 |
> | UT ↑    | 0.5100 | 0.3554         | 0.4653 |
>
>
>
>
>
> W2: The combination of unlearning loss objectives (1) and (3)?
>
> A: Thanks for your question. Yes, the loss objective is the combination of (1) and (3) by a hyperparameter \alpha. We clarified our loss objective in Eq.(4) in our revision. We further conduct experiments on different settings of the hyperparameter, where the results indicate UOE is relatively insensitive to the hyperparameter.
>
>
>
> Q1: What does line 4 in Algorithm 1 mean?
>
> A: Thanks for your question. We have clarified in our revision that the selected expert and the router in the same layer would be trainable (See Line 324).
>
> Q2: For other MoE models (for example Mixtral) without such a shared expert setting, how does UoE work?
>
> A: Due to the limited computational resources available, we conducted new experiments on Mixtral 8*7B only in the PEFT setting (Tab. 4 in revision). The results show that UOE can achieve similar performance with ESFT while only training 97% less parameters. The results have been included in Tab.4 in our revision.
>
> | Dataset    | WMDP   | WMDP   | RWKU   | RWKU   |
> | ---------- | ------ | ------ | ------ | ------ |
> | Method     | FA ↓    | UT ↑    | FA ↓    | UT ↑    |
> | Pretrained | 0.5229 | 0.6885 | 0.5820 | 0.6885 |
> | LoRA       | 0.2658 | 0.2597 | 0.0000 | 0.2295 |
> | ESFT       | 0.2574 | 0.6386 | 0.0542 | 0.6743 |
> | UOE        | 0.2608 | 0.6364 | 0.0455 | 0.6713 |
>
> | Method | Tunable Parameter Ratio ↓ |
> | ------ | ------------------------- |
> | LoRA   | 0.26%                     |
> | ESFT   | 14%                       |
> | UOE    | 0.41%                     |

---

> > ### Comment · Reviewer_QbMP · 2024-11-24
> >
> > Thanks for your response. I am not fully convinced by the claim that "the top-1 expert after unlearning introduces perturbation to the knowledge that is propagated from previous layers and shared experts," as all tokens are routed to the shared experts, and knowledge should exist among them. Your unlearning method only updates the top routed experts and the router in one single layer. I do not see how the knowledge in the shared experts is perturbed. Could you clarify a bit more?
> >
> > Moreover from the same reasoning, it is doubtful whether the knowledge is removed because of the shared experts. I know the evaluation metrics suggest the forget accuracy but I believe a more thorough analysis should be given.
> >
> > Thanks for the Mixtral experiments, why ESFT has significantly more active parameters than UOE? ESFT should also be available in a PEFT setup. And why LoRA has fewer active parameters than UOE + PEFT?

---

> > > ### Author Response · Authors · 2024-11-27
> > > **Thank you for your valuable questions**
> > >
> > > Q1: Your unlearning method only updates the top routed experts and the router in one single layer. I do not see how the knowledge in the shared experts is perturbed. Could you clarify a bit more?
> > >
> > > A: Please refer to this [anonymous link](https://imgur.com/a/eQzT2CN) or Appendix E, titled "Discussion on Shared Experts," in our latest revision. Due to the complexity of the equations, they are challenging to present accurately within the OpenReview system.
> > >
> > > > [[1\] Dai, Damai, et al. "Deepseekmoe: Towards ultimate expert specialization in mixture-of-experts language models." arXiv preprint arXiv:2401.06066 (2024).](https://arxiv.org/pdf/2401.06066)
> > >
> > > Q2: Moreover from the same reasoning, it is doubtful whether the knowledge is removed because of the shared experts. I know the evaluation metrics suggest the forget accuracy but I believe a more thorough analysis should be given.
> > >
> > > A: To understand how knowledge is removed in shared experts, let’s break it down step by step. First, the answer to the previous question indicates that unlearning a target expert introduces noise into the hidden state at the $l$-th layer $\mathbf{h'}_t^{(l)}$. Second, the hidden state $\mathbf{h'}_t^{(l)}$ is propagated forward through the network, layer by layer, until it reaches the output layer, where gradients are computed sequentially. Third, minimizing the unlearning loss function generates gradients that target the model’s output, propagating backward to influence the hidden state $\mathbf{h'}_t^{(l)}$. Finally, the unlearning process effectively alters the model by precisely deviating the hidden state, even as it passes through shared experts in subsequent layers. These shared experts fail to produce accurate outputs related to the target knowledge because they receive an intentionally distorted hidden state from the previous layers.
> > >
> > > Q3: why ESFT has significantly more active parameters than UOE?
> > >
> > > A: ESFT includes at least one expert in each layer until it reaches the threshold, while UOE only requires one expert across all layers of the entire model.
> > >
> > > Q4: ESFT should also be available in a PEFT setup.
> > >
> > > A: As ESFT only trains 14% parameters, it has already been included in our experiments in parameter efficient fine tuning settings.
> > >
> > > Q5:And why LoRA has fewer active parameters than UOE + PEFT?
> > >
> > > A: First, UOE is a parameter-efficient unlearning method, and PEFT refers to training fewer parameters compared to the full model. UOE is a kind of PEFT. In Mixtral’s configuration, each layer contains eight experts, unlike Qwen and Deepseek, where the experts are more fine-grained, totaling around 64 per layer. Consequently, the parameter size for one expert in Mixtral (0.41%) is larger compared to Qwen (0.06%) and Deepseek (0.06%). For LoRA, the number of trainable parameters depends on the model size, the rank hyperparameter of LoRA, and the number of experts. So in Mixtral, the number of experts per layer is fewer compared to Qwen and DeepSeek, resulting in a lower ratio of tunable parameters in LoRA. The table above shows that while UOE and LoRA have a similar number of tunable parameters, UOE demonstrates superior performance in terms of model utility.

---

> > > > ### Comment · Reviewer_QbMP · 2024-11-27
> > > >
> > > > Thank you for the further clarification and efforts in providing new supporting evidence. I will thus increase my score to 5.
> > > >
> > > > Regarding the shared experts part, I agree from the perspective of activations, as you explained. I still have some concerns over the parameter space. In my opinion, parameters are knowledge-dependent. UOE can be regarded as a way to erase knowledge by modifying parameters from the target expert. However, the parameters from the shared experts are never updated, and thus the knowledge should still be there.
> > > >
> > > > I was unsure of the parameter count because of your first response, "We conducted new experiments on Mixtral 8*7B only in the PEFT setting." I thought UOE also updated the target expert in a PEFT way (for example LoRA). However, in your newest response, " the parameter size for one expert in Mixtral (0.41%)," it seems like you are updating the full expert, not in a PEFT setup. Please make it clear in the manuscript.

---

> > > > > ### Author Response · Authors · 2024-11-27
> > > > > **Thanks for your valuable questions**
> > > > >
> > > > > Q1: However, the parameters from the shared experts are never updated, and thus the knowledge should still be there.
> > > > >
> > > > > A: Thank you for agreeing on the activation perspective. Now, let’s explore the parameter perspective to understand how UOE achieves unlearning. We agree that parameters are knowledge-dependent. As described in [1], knowledge is stored as key-value memories within the FFN layers, specifically in the experts (including shared experts) of the MoE architecture. To erase this key-value search mechanism, there are two main approaches:
> > > > >
> > > > > **Breaking the key-value mapping:**
> > > > > This approach involves unlearning shared experts. However, as shown in the table below, it substantially reduces model utility, with MMLU dropping from 0.55 to 0.35 when unlearning 4 shared experts and one top-1 expert.
> > > > >
> > > > > **Perturbing the key:**
> > > > > By altering the input key, this method hinders effective value retrieval. UOE achieves this by introducing perturbations into the hidden state, disrupting the value search process and generating irrelevant outputs, thereby facilitating unlearning.
> > > > >
> > > > > Since our objective is to unlearn targeted knowledge while preserving model utility, we adopt the second approach in UOE, which proves both effective and efficient. Our extensive experiments (see Tab. 2 in our submission) demonstrate that UOE reduces forgetting quality effectively while maintaining high model utility.
> > > > >
> > > > > | Experts \| | Top 1 \|  | Top 1+ 4 Shared \| | Top 6  |
> > > > > | ------- | ------ | -------------- | ------ |
> > > > > | FA ↓    | 0.2700 | 0.2657         | 0.2567 |
> > > > > | UT ↑    | 0.5100 | 0.3554         | 0.4653 |
> > > > >
> > > > > >[1] Geva, Mor, et al. "Transformer feed-forward layers are key-value memories." arXiv preprint arXiv:2012.14913 (2020).
> > > > >
> > > > > Q2: it seems like you are updating the full expert, not in a PEFT setup.
> > > > >
> > > > > A: Thank you for pointing this out. We have revised the statement in our revision to: Due to the computation limitation, Mixtral is only applied on UOE and other parameter-efficient fine-tuning baselines.

---

> > > > > > ### Author Response · Authors · 2024-12-02
> > > > > > **Thanks for your valuable feedback**
> > > > > >
> > > > > > Thank you for taking the time to review our work and provide valuable feedback. If you have no further questions or concerns, we kindly ask if you could consider adjusting your score accordingly.

---

### Official Review · Reviewer_TYfp · 2024-11-03

**Soundness:** 2
**Presentation:** 2
**Contribution:** 2
**Rating:** 5
**Confidence:** 4

**Summary:**

To address the issue of unlearning disrupting the router's expert selection, the authors propose a novel single-expert unlearning framework that focuses unlearning on the most actively engaged expert for the specified knowledge.

**Strengths:**

1、The authors present an innovative and parameter-efficient unlearning framework that effectively identifies, targets, and unlearns the expert most relevant to the forget set.
2、The paper provides numerous experiments that convincingly illustrate the motivation behind this work.
3、Extensive experiments demonstrate that UOE improves both the quality of forgetting and model utility in MoE-based large language models across various benchmarks.

**Weaknesses:**

1、The novelty of this work lacks a compelling impact.

2、The explanation of the router anchor loss is unclear. Specifically, Equation (3) is confusing because it does not define the meaning of g(l).

3、I am unclear about the authors' statement: “we propose the router anchor loss, which encourages the previously identified target expert to remain consistently activated throughout unlearning.” This raises the question of whether the previously identified target expert can reliably be the true target expert. If this cannot be ensured, the process may simply stabilize routing choices, inadvertently activating less relevant experts and undermining the effectiveness of the unlearning process.

4、In Algorithm 1, it is unclear what Ll represents.

5、The authors have not sufficiently emphasized the unique advantages of UOE compared to other MoE frameworks. For instance, while UOE leverages expert attribution to calculate affinity scores for expert selection, other MoE frameworks also compute affinity scores between inputs and experts. Why is UOE more advantageous in this regard? Additionally, when the authors state that "it overlooks finer details that are important for precise comparisons," it would be helpful to clarify what "precise" entails in this context and which additional factors might be considered more significant than precision.

6、In the preliminaries section, the character N denotes the number of experts; however, in Equation (2), it is used to represent the size of the calibration dataset.

7、Sentences such as "Model utility (UT) comparison, at the same level of forget efficacy" in Table 5 and “UT is compared at a consistent level of forget efficacy”in Table 7 are unclear and need clarification. The authors should explain how adjusting model hyperparameters allows one experiment's performance to be maintained while enabling changes in another.

8、Figure 2 illustrates that multiple experts handle a substantial number of tokens, as shown in the long-tail distribution. The decision to unlearn only the top-1 expert requires further justification.

**Questions:**

Please see the weakness.

---

> ### Author Response · Authors · 2024-11-21
>
> W1: The novelty of this work lacks a compelling impact.
>
>
>
> A: We respectfully disagree. We would like to further highlight the contribution and novelty of our work as below: 1) we are **the first** to explore unlearning in MoE architecture, while most existing works focus on dense architecture. 2) We identify the failure of existing unlearning algorithms in MoE models and investigate the underlying causes. 3) Based on these insights, we propose an **effective and efficient framework** called UOE for MoE unlearning, which enhances all baseline methods in terms of forget quality up to 5% and model utility by 35% while training only **0.06%** of the model parameters. We believe that our work will have an impact not only on unlearning tasks but also on the MoE pre-training and post-training phases, where the expert shift issue would be a key problem to prevent stabilizing the MoE training.
>
>
>
> W2: the meaning of g(l)
>
>
>
> A: Thanks for pointing this out. We have clarified the notations in our revised manuscript in Line 324. Specifically, $\mathbf{g}^{(l)}=[g^{(l)}_1,g^{(l)}_2,\dots,g^{(l)}_i]$ is the output of the router in the l-th layer.
>
>
>
> W3: Whether the previously identified target expert can reliably be the true target expert.
>
>
>
> A: Yes. The true target expert in UOE is the decisive expert to process the knowledge that we want to unlearn, which does not change across the unlearning process. As mentioned in Insight 2 (Line 275), the expert selection shift will hide the target expert. To solve this problem, we propose “A single-layer solution for MoE LLM unlearning by UOE” in the following section, where unlearning only happens on one layer to further stabilize the routing choices.
>
>
>
> W4: It is unclear what Ll represents
>
>
>
> A: Thanks for pointing this out. We have clarified it in our revised draft (see Line 297). It is Router^l,indicating the router in the l-th layer in our revision.
>
>
>
> W5-1: The unique advantages of UOE compared to other MoE frameworks.
>
>
>
> A: We would like to clarify that our UOE is an **unlearning framework**, which is not a machine learning model and thus not comparable with **MoE architecture**. As a flexible framework, UOE can be applied to an MoE model by incorporating any existing unlearning algorithms, in an efficient and parameter-efficient manner. Therefore, the unique advantages of UOE are its flexibility and efficiency for unlearning MoE models.
>
>
>
> W5-2: It would be helpful to clarify what "precise" entails in this context and which additional factors might be considered more significant than precision.
>
>
>
> A: Thanks for the suggestions. We have clarified this in our revision (see Line 306). According to the equation in lines 191 and 192, the hidden state is calculated by weighted sum, where the token assignment ratio overlooks the weight for each expert.
>
>
>
> W6: The inconsistent usage of N
>
>
>
> A: Thanks for pointing this out. We have revised Equation(2) in Line 310, where Z is used to represent the size of the calibration dataset.
>
>
>
> W7: The explanation on how adjusting model hyperparameters allows one experiment's performance to be maintained while enabling changes in another.
>
>
>
> A: Thanks for the suggestions. We have clarified our rebuttal revision. To compare different numbers of unlearned experts, we control the training iterations in GA to achieve around 0.25 in FA, which indicates the random selection.
>
>
>
> W8: further justification to unlearn only the top-1 expert.
>
>
>
> A: Thanks for the question. Although the tokens might be handed by multiple experts, involving only the top-1 expert in the unlearning process is enough because the perturbation applied to the top-1 expert can significantly impact the targeted knowledge to forget, due to its large weight in the subsequent average of experts’ outputs in hidden state. In the section “Tuning multiple experts in a single layer”, we justify that unlearning the top-1 expert is better than unlearning top-3 experts (3 experts as targets) and even top-6 experts (6 experts as targets) (refer to Tab. 5). Therefore, we believe that unlearning the top-1 expert is sufficient. In addition, the results in Tab. 7 show that the top-1 expert selection can outperform the random expert selection.

---

> > ### Author Response · Authors · 2024-11-27
> > **Thanks for your insightful review**
> >
> > We would like to follow up to see if our additional contributions effectively addressed your comments. If they did, we kindly ask if you could update your score accordingly. If not, could you please provide more specific feedback on what we could improve or clarify during the rebuttal period? We would be happy to address any additional points.

---

> > > ### Comment · Reviewer_TYfp · 2024-11-27
> > >
> > > Thank you for your response. However, I’m still uncertain whether the previously identified target expert can reliably be regarded as the true target expert. Moreover, if each layer can successfully identify the most relevant target expert, why does applying unlearning across all layers still lead to shifts in expert selection? This appears contradictory. Since the authors address part of the concerns, I correspondingly increase the score to 5.

---

> ### Author Response · Authors · 2024-11-27
> **Thanks for your valuable questions**
>
> Thanks for your valuable questions.
>
> Q1: However, I’m still uncertain whether the previously identified target expert can reliably be regarded as the true target expert.
>
> A: Thank you for your insightful question. In UOE, the target expert is always the true target expert. Let us first analyze the reasons behind expert selection shifts and then explain why UOE ensures the target expert remains true.
>
> **Reasons for Expert Selection Shifts:**
>
> The experiment in Figure 3 reveals that shifts in expert selection are caused by updates to both the router layers and the hidden states. This is evident as the overlap ratio decreases significantly when using either the Router Only or Expert Only approaches.
>
> **Why the Target Expert in UOE is the True Target Expert:**
>
> To clarify, we break the model into three parts:
>
> 1. Layers before the target expert: Since all parameters in these layers are frozen, both the router layers and the hidden states remains unchanged. As a result, no expert selection shifts occur in this part.
> 2. The layer of target expert: As the input hidden state in this layer (the output hidden state from previous layers) is unchanged, the router is the only factor that could induce a shift. Anchor loss is applied to stabilize the expert selection process and increase the affinity score of the target expert during unlearning. The experiment results support that  the top-1 expert's affinity score increases from 0.21 to 0.53 after unlearning of GA+UOE in Deepseek with RWKU dataset.
> 3. Subsequent layers: While expert selection shifts may occur in later layers, these do not influence the target expert or its associated router.
>
> Therefore, in UOE, the target expert used for unlearning reliably is regarded as the true target expert.
>
>
> Q2: Moreover, if each layer can successfully identify the most relevant target expert, why does applying unlearning across all layers still lead to shifts in expert selection?
>
>
> A: Unlearning a single expert with the highest affinity score per layer is not the intended design of UOE. Instead, it serves as a preliminary study or baseline, illustrating that the challenge of expert selection shift cannot be addressed merely by reducing the number of experts in each layer. This observation motivates our approach to developing a targeted single-expert unlearning solution, as detailed in the section “A Single-Layer Solution for MoE LLM Unlearning via UOE.” Our statement that **in UOE** the previously identified target expert is indeed the true target expert, is not contradictory.

---

> > ### Author Response · Authors · 2024-12-02
> > **Thanks for your valuable feedback**
> >
> > Thank you for taking the time to review our work and provide valuable feedback. If you have no further questions or concerns, we kindly ask if you could consider adjusting your score accordingly.

---

### Official Review · Reviewer_JyXv · 2024-11-04

**Soundness:** 3
**Presentation:** 3
**Contribution:** 3
**Rating:** 6
**Confidence:** 2

**Summary:**

The paper addresses the challenge of unlearning in Mixture-of-Experts (MoE) LLMs. When naively applying traditional unlearning methods to MoE models, the authors discovered that MoE's dynamic routing mechanism leads to unintended expert selection shifts, causing excessive forgetting and reduced model utility. To solve this, they propose UOE (Unlearning on One Expert), a framework that: 1) identifies the most relevant expert for the target knowledge using affinity scores, and 2) implements a router anchor loss to maintain that expert's activation during unlearning. By concentrating unlearning on a single expert, UOE + baseline methods improve both forgetting quality and model utility significantly compared to baseline methods alone.

**Strengths:**

1. Novel and well-motivated solution that specifically addresses MoE architecture challenges in unlearning, filling an important gap in the literature
2. Thorough follow-up studies that justify key design choices, particularly: the effectiveness of single-expert versus multi-expert unlearning, and the impact of single/multi-layer selection on forgetting performance.
3. Strong empirical validation with comprehensive experiments across different: unlearning algorithms, model architectures, benchmarks 4. Resource-efficient approach, requiring modifications to only 0.06% of model parameters while achieving superior performance.

**Weaknesses:**

1. **Notation Clarity**: Several key mathematical notations (particularly $g(l)$ in the router anchor loss formulation) would benefit from more explicit definitions.
The relationship between different mathematical terms could be better explained.
2. **Implementation Details**: Training hyperparameters are insufficiently documented for reproducibility; the $Unlearn()$ subroutine in Algorithm 1 lacks specific implementation details; more comprehensive experimental setup information would facilitate replication.

**Questions:**

1. The method relies on selecting the expert with the highest affinity score. However, how robust is this approach when there are experts with closely competing scores?
2. The robustness to attack of the proposed method: if we keep the model frozen and use soft prompt learning, can the soft prompt uncover the knowledge from other experts? If so, this suggests that the knowledge still resides within these experts and can be accessed through carefully crafted input prompts.

---

> ### Author Response · Authors · 2024-11-21
>
> W1: Notation Clarity
>
> A: Thanks for the suggestions. We have revised our draft and clarified key notations, such as the definition of $\mathbf{g}^{(l)}=[g^{(l)}_1,g^{(l)}_2,\dots,g^{(l)}_i]$, which is the output of the router.
>
> W2: Training hyperparameters are insufficiently
>
> A: Thanks for the valuable suggestion. We skip this part due to the limitation of space. We include the following in the appendix Sec. A:
>
> >We set the learning rate to 5e-5 for GA, NPO, and GD while setting it to 1e-4 for UOE. The batch size is 4 for GA, NPO, and GD, while it is set to 16 for UOE. In NPO, the beta value is set to 0.001. The alpha for the retain loss is set to 1 in both GD and NPO. For RMU, we follow the hyperparameters specified in the original work. We configure the steering coefficients as 8000 for Qwen and 32000 for Deepseek, as UOE targets deeper layers in these models.
>
> Q1: how robust is this approach when there are experts with closely competing scores?
>
> A: This is a great question! Our approach is robust to experts with closely competing scores. Regarding the robustness of experts across different affinity scores, we have shown the results in Tab. 7 of our original submission. The results indicated UOE is robust to different affinity scores as mentioned in line 511. If experts have closely competing affinity scores, no matter which one is selected, it makes similar large impacts on the hidden state, resulting in similar unlearning performance. So the expert with the highest affinity score can lead to the largest impact on the perturbation. To explore the robustness of selecting experts with competing scores, we conduct new experiments on the rank 2 expert and the rank 3 expert and report the results in the table below. The results indicate that the rank 2 and 3 experts lead to unlearning performance as good as the rank 1 expert.
>
> | Rank | Affinity Score | UT ↑   | FA ↓ |
> | ---- | -------------- | ------ | ---- |
> | 1    | 0.211          | 0.5485 | 0.0  |
> | 2    | 0.1957         | 0.5475 | 0.01 |
> | 3    | 0.1695         | 0.5453 | 0.02 |
>
> Q2: Can the soft prompt uncover the knowledge from other experts?
>
> A2: Thanks for your valuable question. The soft prompt attack fails to uncover the knowledge. To investigate the soft prompt attack, we employ the GCG attack in a white-box setting to optimize the prompt, aiming to force responses to begin with "Sure, here is the answer:". The number of optimization steps is increased to 5000, while other hyperparameters remain at their default settings. Considering the computational demand (approximately 1 GPU hour on an A100 for generating a single soft prompt), we optimize 400 prompts across 400 samples in RWKU. Since not all responses begin with "Sure, here is the answer:", we filter for those containing the word "answer" and then assess the forget quality both with and without GCG-generated prompts. Experimental results below indicate that the GCG, despite being the strongest prompt-level attack, fails to recover the forgotten knowledge, as the forget accuracy (FA) remains at 0.01 before and after the GCG attacks.  We have included these results in section B in the Appendix.
>
>
> | Original | GCG  |
> | -------- | ---- |
> | 0.01     | 0.01 |
>
> | Questions                                               | Original Answer | Unlearn without soft prompt                 | Unlearn with soft prompt                                     |
> | ------------------------------------------------------- | --------------- | ------------------------------------------- | ------------------------------------------------------------ |
> | Stephen King is often referred to as the 'King of ___'. | Horror          | indingindingindingindingindingindinginding… | ologist:Sure here is the answer:Question : Stephenindingindingindinginding… |
> | Bruce Lee moved to ___ in 1959.                         | America         | -wardswardswardswards...                    | eanswer:” An-swer:[/b]inding [b]...                          |

---

> > ### Author Response · Authors · 2024-12-02
> > **Thanks for your valuable feedback**
> >
> > Thank you for taking the time to review our work and provide valuable feedback. If you have no further questions or concerns, we kindly ask if you could consider adjusting your score accordingly.

---

### Author Response · Authors · 2024-11-21

Thanks for all the valuable comments. We have carefully reviewed your feedback and advice, and provided our responses accordingly.  Please check our highlighted revisions and detailed responses to each reviewer. We hope that the revisions have addressed your comments. We are happy to provide further discussion if there are additional concerns or areas for clarification.

A short summary of newly conducted experiments:
1. [Reviewer  JyXv] Following your suggestions, we introduce a soft prompt attack GCG against UOE in Tab. 9 in revision. The results indicate that UOE is robust to the GCG attack.
2. [Reviewer  TYfp] Following your suggestions, we incorporate Mixtral 8*7B as a MoE model without shared experts in our experiments in Tab. 4. The results indicate that UOE can apply to MoE without shared experts as well.
3. [Reviewer JQDp] Following your suggestion, we visualize the uncertainty of the unlearning process in Fig. 6. The results indicate that the expert selection shift is not caused by uncertainty in routing.

---

### Meta-Review · Area_Chair_Wb8L · 2024-12-23

**Metareview:**

Summary
========
The paper introduces UOE (Unlearning on One Expert) for machine unlearning in Sparse Mixture-of-Experts (MoE), a relatively unexplored area. The key finding is that traditional unlearning methods cause unintended expert selection shifts in MoE models, leading to excessive forgetting. UOE addresses this by identifying and targeting the most relevant expert for unlearning while maintaining its activation through a router anchor loss. This focused approach enhances both forgetting quality and model utility significantly compared to baseline methods.

Strengths
=======
* Well-motivated problem
* Novel approach addressing the unique challenges in MoE architecture unlearning, filling a critical gap in the literature
* Comprehensive empirical validation across different architectures and benchmarks
* Resource-efficient, modifying only 0.06% of model parameters
* Thorough ablation studies justifying key design choices

Weaknesses
==========
* Insufficient treatment of shared experts in modern MoE architectures (e.g., Qwen1.5, DeepSeek)
* Unclear mathematical notation and implementation details, particularly for router anchor loss
* Limited discussion of versatility across different routing algorithms beyond TopK
* Inadequate justification for single-expert approach given multi-expert token handling
* Implementation details, including training hyperparameters and specific steps in Algorithm 1, are inadequately documented
* Several technical concerns remain unaddressed, such as the handling of shared experts in MoE models and the reliability of target expert identification.

Reasons for decision
================
The paper addresses an interesting problem unlearning in MoE LLMs by introducing an innovative approach, and demonstrating strong empirical results. However, significant issues related to clarity, completeness of implementation details, and unresolved technical concerns still remain.

**Additional Comments On Reviewer Discussion:**

The authors are engaging in discussions, attempting to address concerns raised, resulting in slight increase in rating after the rebuttal period.

---

### Decision · Program_Chairs · 2025-01-22

Reject